

# Parallel path detection for fraudulent accounts in banks based on graph analysis

Zuxi Chen[1,2], ShiFan Zhang[1,2], XianLi Zeng[3], Meng Mei[4], Xiangyu Luo[1,2] and Lixiao Zheng[1,2]

[1] Huaqiao University, Fujian, China
[2] Xiamen Key Laboratory of Data Security and Blockchain Technology, Xiamen, China
[3] Guilin University of Electronic Technology, Guangxi, China
[4] Tongji University, Shanghai, China

## ABSTRACT

This article presents a novel parallel path detection algorithm for identifying suspicious fraudulent accounts in large-scale banking transaction graphs. The proposed algorithm is based on a three-step approach that involves constructing a directed graph, shrinking strongly connected components, and using a parallel depth-first search algorithm to mark potentially fraudulent accounts. The algorithm is designed to fully exploit CPU resources and handle large-scale graphs with exponential growth. The performance of the algorithm is evaluated on various datasets and compared with serial time baselines. The results demonstrate that our approach achieves high performance and scalability on multi-core processors, making it a promising solution for detecting suspicious accounts and preventing money laundering schemes in the banking industry. Overall, our work contributes to the ongoing efforts to combat financial fraud and promote financial stability in the banking sector.

## INTRODUCTION

In the financial sector, digitalization has swept the banking industry. For decades, banks have upgraded their infrastructure and services. Presently, customers can monitor and control their financial tasks in real time through a considerable number of tools provided by banks (*Schmidt, Drews & Schirmer, 2017*; *Indriasari et al., 2022*). However, as a service provider, banks are able to collect data about financial activities of customers, including the flow of funds between accounts (*Oral et al., 2020*). With the help of the extracted information, banks can offer personalized products, improve the quality of service and the profit of banking system, or can even detect and prevent illegal fraud activities of individuals or companies (*Cheng & Feng, 2021*). For the banking technology department, the biggest challenge is the extraction of the key information from massive activity data.

Based on past fraud activities and money laundering cases, there are some common features that some customers and other customers may have frequent large amount of money transactions or unusually complex transfer paths between customers and customers (*Isa et al., 2015*). In this research we focus on the transaction paths between two

Corresponding author
XianLi Zeng, sofacoder@163.com

accounts and introduce a method for path detection based on parallel depth-first algorithm. Different fraud techniques can be connected to the transaction path such as if an account flows through multiple paths to another account, there may be a transfer of black money. The accuracy of financial regulation on bank accounts can be improved by risk-labeling anomalous accounts in transaction paths (*Hilal, Gadsden & Yawney, 2022*).

In real-world banking scenarios, graphs can be very large and grow exponentially over time, posing significant challenges for graph analysis and processing (*Shabbir et al., 2022*). For example, detecting paths between accounts in a transaction graph can help identify fraudulent activities and prevent money laundering. However, such path detection tasks require substantial memory and computational resources, which cannot be efficiently utilized by single-threaded algorithms. Therefore, there is a need for parallel algorithms that can fully exploit CPU resources and handle large-scale graphs with exponential growth (*Lucas & Sackrowit, 1989*). In this article, we present a novel parallel path detection algorithm based on depth-first search (DFS), which can achieve high performance and scalability on multi-core processors.

We present a novel three-step approach to detect suspicious fraudulent accounts in banks based on graph analysis. First, we construct a directed graph from the bank's transaction records. We use the multi-core on-the-fly strongly connected component (SCC) algorithm, which is a parallel and linear-time algorithm for finding all strongly connected components (SCCs) (*Bloemen, Laarman & van de Pol, 2016*; *Bloemen & van de Po, 2016*; *Bloemen, 2015*). Second, we shrink each SCC into a single vertex and remove any self-loops, obtaining a new directed graph that is acyclic. Third, we use a parallel depth-first search algorithm to mark all vertices that lie on any path from an Outflow accounts to inflow accounts. These marked vertices correspond to potential fraudulent accounts that are involved in money laundering schemes.

The structure of the article is the following. After the 'Introduction' first we give an overview of 'Related Work', then we declare some basic concepts and definitions that are essential for understanding our approach in the 'Preliminaries' section. In 'Preliminaries', we outline the 'General Idea of Our Algorithm' and explain how it differs from existing methods. In the 'Implementation of Our Algorithm' section, we describe our algorithm in detail and analyze its complexity and correctness. We then evaluate our algorithm on various datasets and compare it with serial time baselines in the 'Evaluation' section. Finally, we summarize our contributions and discuss future directions in the 'Conclusion' section.

## RELATED WORK

Fraud detection in the banking sector is a critically important task aimed at safeguarding the financial assets of both banks and customers from fraudulent activities. In recent years, a multitude of methods have emerged for detecting fraudulent accounts in banks, and we have summarized the primary approaches in Table 1.

In the banking sector, machine learning (*Lv et al., 2019*; *Hashemi, Mirtaheri & Shamsi, 2021*; *Patil, Nemade & Soni, 2018*) has emerged as a potent tool for detecting fraudulent accounts. Notably, graph neural networks (*Zeng & Tang, 2021*; *Xiang et al., 2023*) excel in

**Table 1 Summary of related work.**

| Technical solution | Advantages | Limitations |
|---|---|---|
| Residual layered CARE-GNN (RLC-GNN) | This algorithm achieves state-of-the-art results in fraud detection tasks by addressing relation camouflages and feature camouflages. | Deep RLC-GNN models may face overfitting issues. |
| Semi-supervised graph neural network for fraud detection (GTAN) | This method captures associations between temporal transactions and learns representations of transactions, aiding in identifying fraud patterns, even with very few labeled data. | Building temporal transaction graphs and handling large-scale transaction data may require extensive data engineering. |
| Community detection algorithm | This algorithm helps identify patterns that may lead to fraudulent events, enhancing the accuracy of fraud detection. | Community detection algorithms are often sensitive to noise, necessitating data preprocessing to reduce noise. |
| Big data clustering technique and customer behavior indicators | This approach detects and prevents potential fraudulent activities in customer banking transactions in the fastest possible time. | This method requires a large volume of high-quality data to perform optimally. |
| Graph attention network (GAT) | This tool employs a graph attention network, combined with social network metrics as node features, to better capture the roles and patterns of accounts in money laundering activities, resulting in more accurate money laundering detection. | Graph attention networks are relatively complex and require more computational resources and time for training and usage. |
| Decision tree-based fraud detection mechanism by analyzing uncertain data | By selecting features and using decision tree algorithms, this aids banks in better identifying potential fraudulent transactions. Decision trees can facilitate fraud detection in scenarios that require rapid responses, reducing processing delays. | Decision trees may be limiting when dealing with large datasets and complex problems, especially when handling numeric data that could lead to the generation of extensive trees that are challenging to analyze. |

terms of detection accuracy, thanks to their ability to uncover intricate relationships and patterns within vast transaction datasets. Graph attention networks (*Sheu & Li, 2021*), excel at capturing account relationships in social networks and dependencies among nodes in transaction graphs. However, it is inevitable that machine learning methods require more data and feature engineering to effectively capture anomalous patterns within network structures, necessitating additional computational resources and time for training and implementation (*Bao, Hilary & Ke, 2021*; *Erdogan et al., 2020*). In banking transaction data, fraudsters may engage in fraudulent activities with specific patterns or social connections. Community detection algorithms (*Sarma et al., 2023*) aid in identifying these latent patterns, enhancing the precision of fraud detection. But these algorithms prove sensitive to noise in sparse graphs. Big data-driven approaches (*Kian & Obaid, 2022*; *Bănărescu, 2015*) find numerous applications in fraud detection, often delivering rapid results, but they necessitate high-quality data. Decision trees (*Khare & Viswanathan, 2020*) are employed in fraud detection as well. These methods exhibit high responsiveness but can pose challenges when it comes to analyzing extensive data and complex issues.

The relationships between bank accounts typically resemble those of graph nodes. While graph algorithms provide a more direct means of capturing relationships within graph networks, they often face challenges when dealing with large-scale graphs. In recent years, numerous parallel graph algorithms have emerged, and a distributed search approach (*Hao, Yuan & Zhang, 2021*) based on graph partitioning has introduced a novel

solution for parallel searching. Previously, it was considered difficult to parallelize dictionary or ordered DFS (*Reif, 1985*). However, parallel randomized depth-first search (DFS) (*Bloemen, Laarman & van de Pol, 2016*; *Evangelista, Petrucci & Youcef, 2011*) has offered a fresh perspective. Significant progress has been made in the parallelization of directed acyclic graphs (*Naumov, Vrielink & Garland, 2017*), and parallel approaches to finding SCCs have demonstrated promising results in model checking (*Bloemen & van de Po, 2016*; *Laarman, 2014*). In terms of practical applications, parallel DFS has already been employed in train rescheduling (*Josyula, Krasemann & Lundberg, 2021*).

## PRELIMINARIES

Let $G = (V,E)$ denote a unweighted directed graph, where $V(G)$ is the set of vertices and $E(G)$ is a set of directed edges. For a vertex $v \in V(G)$, we define $N(v)$ as the set of neighbors of $v$ and $N_{next}(v)$ as a next neighbor of $v$. We transform $G$ into a directed acyclic graph by collapsing each strongly connected component (SCC) into a single node $w$. We denote by $F(v)$ the node representing the SCC that contains $v$ ($F(v) = w$), by $S(w)$ the set of vertices in the SCC, by $N_{random}(w)$ a random representative neighbor of $w$. Our goal is to identify all possible accounts that may participate in the transfer of funds between an outflow account and an inflow account. We call the outflow account the start node and the inflow account the target node.

### Problem statement

Given a directed graph, a start node $s$ and a target node $t$, enumerate all nodes that are possibly traversed between $s$ and $t$.

### Graph storage

We used an adjacency list as the data structure for the directed graph in this article. In order to minimize memory usage, we implemented a node deduplication strategy for graph nodes, ensuring that each node occupies only one storage space. Additionally, we employed a layered storage strategy, allocating new storage layers only when the capacity is insufficient. Each storage layer consists of an array of node pointers, and nodes are located within this array using both the storage layer and an internal offset. During the construction of the adjacency list, we extended this optimization to the neighboring nodes of each node. We only stored node pointers for neighboring nodes, thereby avoiding redundant storage of a node's data in memory. This approach effectively prevented memory waste resulting from multiple copies of the same node when accessed by different threads during parallelization.

### A sequential DFS algorithm

To introduce our algorithm, we first review the conventional serial algorithm that uses depth-first search to enumerate all nodes between $s$ and $t$. The sequential DFS algorithm explores each branch of the graph as far as possible from the start node until it reaches a dead end, then backtracks to another branch (*Grossi, Marino & Versari, 2018*; *Peng et al., 2019*; *Rizzi, Sacomoto & Sagot, 2014*). When one of the neighbours of the current node can reach the target node, all nodes within the same SCC are marked as pathnodes. Non-

| Algorithm 1 Finding all path nodes between two vertices in a directed graph. |
| --- |

**Input:** The Graph stored in an adjacency list $G$, The start node $s$, The target node $t$

**Output:** a list of path nodes from $s$ to $t$

```
1   visited ← {false};
2   pathNodes ← {t};
3   stack.push(s);
4   visited[s] ← true;
5   while stack is not empty do
6       v ← stack.top;
7       If N(v) have all been traversed then
8           if N(v) ∩ pathNodes! = NULL then
9               pathNodes.add(S(F(v)))
10          end
11          stack.pop;
12          visited[v] ← false;
13      else
14          u ← N_next(v);
15          if visited[u] = false then
16              stack.push(u);
17              visited[u] ← true;
18          end
19  end
```

recursive DFS has the same functionality and applications as recursive DFS, but it avoids the risk of stack overflow caused by deep recursion. Let's take a look at the implementation of the idea we've described in Algorithm 1 (non-recursive):

We begin by initializing some variables in lines 1–4: a visited array with only the start node set to true and the rest to false; a *pathnodes* array with only the target node; and a recursive stack with only the start node. We then perform DFS while the stack is non-empty. In lines 8–12, when we have explored all neighbours of node $v$, we backtrack and examine whether any of them belongs to *pathnodes*. If yes, it implies that node $v$ can reach the target node, as can all nodes in its SCC. Hence, we add all nodes in the current SCC to *pathnodes*. After this check, we pop node $v$ from the stack and mark it as unvisited. In Line 14, we take the next neighbour of node $v$ as node $u$. To avoid infinite loops due to cycles in the graph, we verify whether node $u$ has been visited in line 15. If not, we proceed with DFS from node $u$.

The main challenge for parallelizing single-threaded depth-first algorithms is the sequential nature of forward exploration, which makes later nodes depend on previous results. When one thread searches along a path, other parts of the graph remain

underutilized (*Stone & Sipala, 1986*; *Zhang, 2010*). Our solution is to use multiple threads to explore different paths simultaneously, while sharing information among them.

## GENERAL IDEA OF OUR ALGORITHM

Detecting suspicious fraudulent bank accounts requires a high degree of time sensitivity. Rapidly assessing and identifying potentially fraudulent accounts can mitigate risks and losses. Moreover, it enables the tracking of fund flow, as money laundering activities typically involve complex fund transfers among multiple accounts. Timely identification of suspicious accounts can assist in tracing and understanding the entire process of money laundering, unveiling related criminal networks. Most importantly, a bank's anti-money laundering strategy is only effective when money laundering activities are promptly recognized and halted. Delays in detecting suspicious accounts could provide money launderers with the opportunity to complete their activities, thereby increasing potential risks. Consequently, we have optimized the time required for detecting suspicious accounts through parallelized DFS.

The presence of circles between accounts is regarded as sensitive information in the financial domain, as detecting the existence of such circles can serve as an indicator of potential money laundering activities. Money launderers often employ multiple transfers, effectively cleansing funds by moving them through various accounts before eventually returning them to their original source in an attempt to obfuscate the funds' origin. Additionally, cyber fraudsters may utilize frequent fund cycles and repeated transfers to obscure their fraudulent activities. As a result, our algorithm not only focuses on tracking the flow of funds between accounts but also places significant emphasis on identifying circles within these pathways.

Based on the analysis in 'Preliminaries', we propose a novel concurrent data structure that is tailored for enumerating path nodes in parallel. Our data structure aims to achieve four objectives: (1) parallelism, it should exploit thread resources efficiently; (2) pruning capability, it should avoid redundant visits to the same node by different threads, reducing unnecessary computation and improving enumeration performance; (3) memory consumption, it should minimize the memory usage per node; (4) load balancing, it should distribute the workload evenly among threads and allow random access to neighboring nodes.

We designed a concurrent shared data structure for our algorithm, as shown in Fig. 1. The node data structure consists of three parts: an *isDone* flag, an *isReachable* flag and a dynamic array of pointers to neighboring nodes. The *isDone* flag is crucial for ensuring the parallelism and pruning ability of our algorithm. It indicates whether a thread has finished traversing the neighbors of a node. If another thread encounters a node with an *isDone* flag set to true, it will not visit it again. Otherwise, the thread will process the node. The *isReachable* flag indicates whether the target node can be reached through the current node. In the backtracking process, we use the *isReachable* values of the neighbors to determine whether the current node is reachable. To reduce memory consumption, we use an array of neighbor node pointers to store the addresses of each node in memory. The dynamic array can adapt well to different numbers of neighbors for each node.

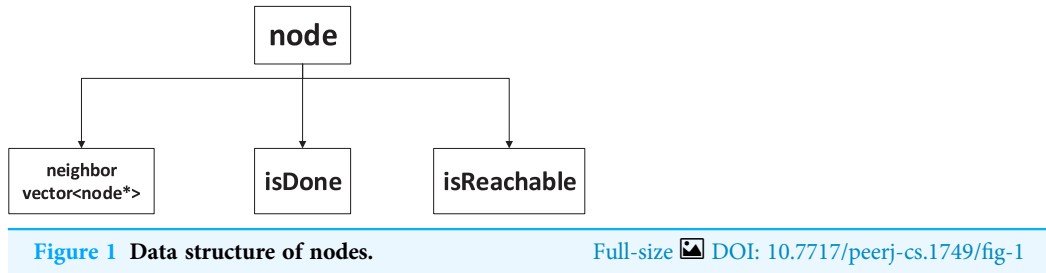

**Figure 1  Data structure of nodes.**               

The purpose of designing this data structure is to enable multiple threads to cooperate with each other and to distribute the paths explored by each thread evenly throughout the graph. This way, we can improve the efficiency and scalability of our algorithm and avoid redundant computations.

In Algorithm 1, the sequential DFS algorithm uses a *visited* array to detect cycles in a graph. If it finds an adjacent vertex that has already been visited during traversal, it indicates a cycle. However, in a multi-threaded environment, using a *visited* array would cause one thread to occupy a node, preventing other threads from exploring subsequent node through that node. This would greatly reduce parallelism. If there are large cycles in the graph, it would make the algorithm's performance degrade even faster.

To solve this problem, before running our algorithm, we use the multi-core on-the-fly SCC algorithm (*Bloemen, Laarman & van de Pol, 2016*; *Bloemen & van de Po, 2016*; *Bloemen, 2015*). This algorithm can return SCCs on-the-fly while traversing or generating the graph and has linear time complexity. An example is the most effective way to explain the strategy. Figure 2 shows a specific scenario where two workers(*blue, red*) can explore a SCC more efficiently when compared to a sequential version.

Starting from node $a$, both threads randomly select an adjacent node to explore the graph: the blue thread goes to node $b$ and the red thread goes to node $d$ (Fig. 2A). Assuming that the red thread has already explored the path $a \rightarrow d \rightarrow e \rightarrow a$ and thus discovered a cycle $\{a, e, d\}$, while the blue thread has explored the path $a \rightarrow b \rightarrow c \rightarrow b$ and discovered a cycle $\{b, c\}$. The SCCs discovered by the two threads have been marked in Fig. 2B. Now, the blue thread continues to visit the unvisited neighboring node $d$ of node $c$ (Fig. 2C). Without exchanging information between the threads, the blue thread needs to continue exploring the path $d \rightarrow e \rightarrow a$ to discover the cycle $\{a, e, d, c, b, a\}$. However, as shown in Fig. 2D, the red thread knows that nodes $d$ and a part of the same SCC. If the two threads exchange node information correctly, the blue thread can determine that $c \rightarrow d$ is equivalent to $c \rightarrow *a$, since node $a$ also belongs to the exploration path of the red thread. Finally, the red thread adds nodes $b$ and $c$ to the red SCC ($\{a, e, d\}$). Therefore, the complete SCC is $\{a, b, c, d, e\}$.

We apply this algorithm to find all SCCs in the graph and aggregate all vertices in an SCC to a representative vertex. This way, we can effectively eliminate cycles in the graph and improve parallelism and efficiency of multi-threaded DFS. Figure 3 demonstrates how to convert a directed graph to a directed acyclic graph. In the strongly connected component $a, b, c$, node $c$ is chosen as the representative node. For node $a$, $F(a) = \{c\}$ and

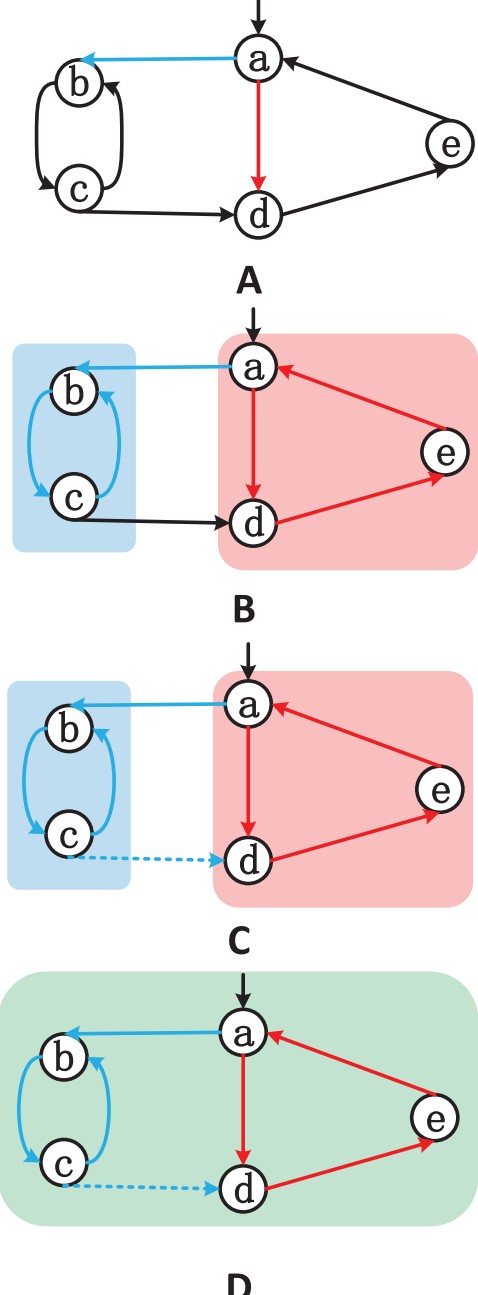

**Figure 2** **(A–D) Example of two threads discovering SCCs.**

$S(F(a)) = \{a, b, c\}$. Node $c$ inherits all the node relations of $a, b, c$, that is, the original $b \rightarrow d$ and $a \rightarrow d$ become $c \rightarrow d$ after the conversion.

Sequential depth-first search is a recursive algorithm that keeps exploring the vertices of the graph in one direction until it hits a dead end. When we convert bank transaction records into a directed graph, the order of visiting neighbor vertices in the adjacency list is fixed for each vertex. In Algorithm 1, line 14, the next vertex to explore can only be $N_{next}$,

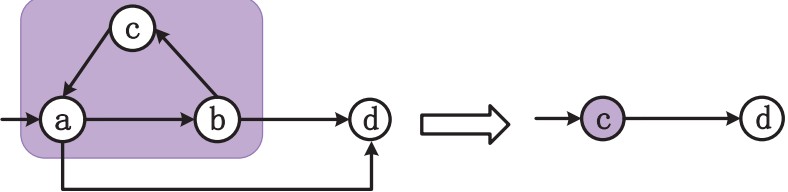

**Figure 3** The process of converting to a directed acyclic graph.

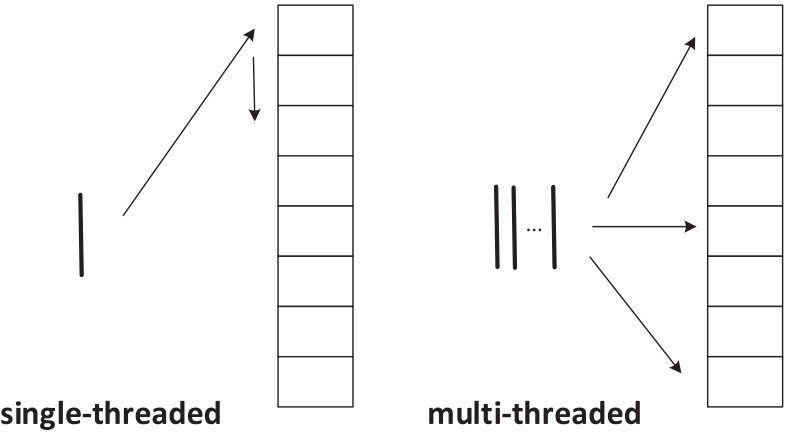

**single-threaded**          **multi-threaded**

**Figure 4** Single-threaded *vs* multi-threaded execution.

which is very unfavorable for parallelization. To overcome this limitation, we introduce randomness in our algorithm. For each thread, we randomly select one of the neighbor vertices as the next vertex to explore in each iteration. This can guide threads to explore different directions in the graph and greatly improve the parallelism of the algorithm. Figure 4 below shows the difference between multi-threaded and single-threaded methods of accessing neighboring nodes.

We designed an algorithm that allows each thread to collaboratively update node states in a shared data structure and jointly perform graph search. The general idea of our algorithm is best explained using the example from Fig. 5. In Fig. 5, two threads (or workers), which we call *red* and *blue*, start their search from the start node *a*. Here, node *e* is the target node, and the goal is to find all nodes between the start node and the target node. During the process of two threads traversing the graph, if one thread completes the traversal of all neighboring nodes of a particular node, that node is marked as *Done*, indicating that other threads no longer need to traverse it. In Fig. 5, we use red or blue to fill nodes, representing vertices marked as *Done* by two different threads. The *pathnode* array in the bottom right corner stores nodes currently marked as *Reachable*, meaning nodes that can potentially reach the target node. *Reachable* in the current stage, meaning nodes that are potentially reachable to the target node.

In Fig. 5A, we illustrate how our parallel graph exploration algorithm starts. Both red and blue threads start from node *a* and randomly select one of its neighbors to explore. For example, if the blue thread selects node *d*, it will proceed along the path $a \rightarrow d \rightarrow c$.

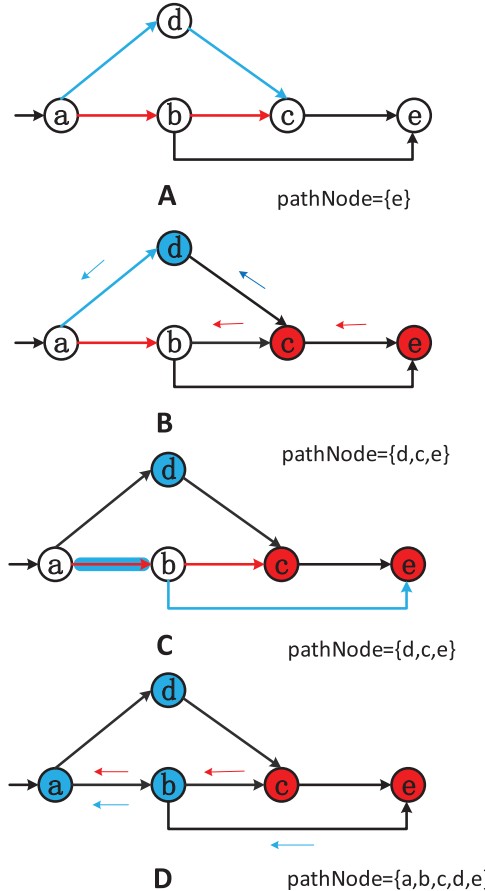

**Figure 5 (A–D) The process of parallel depth-first search.**

Similarly, after two random selections at nodes $b$ and $d$, the red thread will follow the path $a \to b \to c$. In this stage, the *pathnode* array contains only the target node, which is node $e$. This random selection strategy enables sufficient parallelism for our algorithm.

In Fig. 5B, the red thread explores forward along the path $a \to b \to c \to e$. Node $e$, serving as the target node, doesn't have any neighboring nodes by default, so it is marked as *Done*. When the red thread backtracks from the reachable node $e$ to node $c$, it marks node $c$ as a reachable node and adds it to the *pathnode*. At the same time, the blue thread advances along the path $a \to d \to c$. However, when it encounters node $c$, it realizes that node $c$ has already been marked as *Done* by the red thread and is a reachable node. Consequently, the blue thread ceases further exploration. Upon backtracking to node $d$, it marks node $d$ as a reachable node and adds it to the *pathnode*. After confirming that there are no other neighboring nodes left to explore for node $d$, it is then marked as *Done*.

Now consider what happens in Fig. 5C. When the blue thread backtracks to node $a$, it continues to explore along the unvisited node $b$. At node $b$, the blue thread and the red thread have different choices. The blue thread chooses to explore towards ndoe $e$. In our random process, the two threads may sometimes choose the same path, but they will

diverge again after exploring a segment of the path. This allows the threads to fully explore different parts of the graph.

In Fig. 5D, after exploring node *e*, blue thread backtracks to node *b* and finds that all remaining neighbours of b have been visited by red thread. Therefore, blue thread marks node *b* as done and inherits the reachability of node *e*, adding node *b* to the *pathnode*. Similarly, when backtracking to node *a*, blue thread also marks nodes *a* as done and reachable, and it is included in the *pathnode*. It can be observed that due to the randomness of numbers, many identical nodes may appear in the path stacks of both threads. In this case, whichever thread processes a node faster will determine its status, and the other thread can simply pop it out of the stack when encountering it. This reflects the flexibility of our algorithm.

From the above parallel example, it is evident that the red and blue threads effectively share the task of traversing the graph. The blue thread primarily handles nodes a, b, and d, while the red thread focuses on nodes c and e. This efficient task distribution significantly reduces the program's execution time. In our comprehensive algorithm, we aggregate highly suspicious loops into single nodes, thereby enhancing the parallelism of traditional DFS. We introduce an element of randomness to guide each thread to different areas of the graph. Through the use of *Done* and *Reachable* markers, we prevent multiple threads from redundantly exploring the same graph segments. This strategy effectively achieves parallel DFS for identifying suspicious accounts in networked graphs.

After identifying these suspicious bank accounts, financial institutions must establish time and monetary thresholds based on the specific context. If transaction records reveal significant fund transfers between the fund output and fund input accounts within a short timeframe, there is a substantial likelihood that these transactions are intended to conceal the illicit source of funds through multiple account transfers. This may involve money laundering, online fraud, and other illicit activities. Following verification by the financial institution, these accounts will be officially designated as fraudulent.

This practice enables financial institutions to more accurately detect potential fraudulent activities, particularly those attempting to obscure their criminal activities through complex transactions. Auditors or risk management professionals within financial institutions can more effectively monitor risks in fund transactions by setting time and amount thresholds, thereby enhancing anti-fraud and anti-money laundering measures, ensuring transparency and security within the financial system.

## IMPLEMENTATION OF OUR ALGORITHM

Before running our algorithm, we optimized the graph using the multi-core on-the-fly SCC algorithm (*Bloemen, Laarman & van de Pol, 2016*; *Bloemen & van de Po, 2016*; *Bloemen, 2015*), as shown in Algorithm 2.

To ensure the correctness of the parallel algorithm, each line in Algorithm 2 is an atomic operation. Each thread has its own local search stack stack and the global sets *S*, Dead and Completed are initialized in lines 26–28. Dead means that an SCC has been fully explored, and Completed means that a node has been fully explored. In line 3, each iteration starts

**Algorithm 2  UFSCC algorithm.**

```
1   Function UFSCC (v)
2       stack.push(v);
3       while v′ = S(v) \ Completed do
4           foreach w ∈ Random(N(v)) do
5               if w ∈ Dead then
6                   continue;
7               end
8               else If ∄w′ ∈ stack : w ∈ S(w′) then
9                   UFSCC (w);
10              end
11              else
12                  while S(w) ≠ S(v) do
13                      Unite(S,stack.top(),stack.pop());
14                  end
15              end
16          end
17          Completed = Completed ∪ {v′};
18      end
19      if S(v) ⊄ Dead then
20          Dead = Dead ∪ S(v);
21          reportSCC;
22      end
23      if v = stack.top() then
24          stack.pop();
25      end
26      foreach thread do
27          stack ← ∅;
28          Completed = Dead ← ∅;
29          S(v) = v : v ∈ V(G);
30          UFSCC{w};
31      end
```

from a node $v$ in S that is not completed and searches for an SCC. For each neighbour $w$ of $v$, lines 4–16 describe the three possible states and actions of $w$:

- If $w$ is dead (line 5), it means that $w$ is part of a completed SCC and can be ignored.
- If $w$ is not in the local stack stack and is not part of another SCC from stack (line 8), it means that $w$ has not been visited by any thread and its S(w) contains nodes that need to be further explored, so the UFSCC function is recursively called on $w$.

- If *w* is in the local stack (lines 12–15), it forms a cycle and all nodes on the cycle are in an SCC. We assume that partial SCCs adhere to the strong connectivity property and that the search stack sufficiently captures them.

After exploring all neighbours of *w*, *w* is added to Completed. After detecting a cycle, if *v* is at the top of the local stack, a completed SCC is reported and marked as Dead, indicating that *v* cannot be merged with any node (lines 19–20).

We constructed a directed graph from bank transaction data and preprocessed it using the multi-core on-the-fly SCC algorithm. Based on this, we applied our parallel depth-first search algorithm to find all possible paths from the start node to the target node that visit every node in the graph. We present the pseudocode of our implementation in Algorithm 3.

We define four possible states for each node in the graph, based on two boolean flags: *isDone* and *isReachable*. The flag *isDone* indicates whether all the neighbors of the node have been explored or not. The flag *isReachable* indicates whether the node belongs to a SCC that can reach the target node or not. The four states are:

- *isDone* = false and *isReachable* = false: This state means that the node has some unexplored neighbors, and it cannot reach the target node through any of its current paths.
- *isDone* = false and *isReachable* = true: This state means that the node has some unexplored neighbors, but it belongs to a reachable SCC, so it can potentially reach the target node through some other nodes in its SCC.
- *isDone* = true and *isReachable* = false: This state means that all the neighbors of the node have been explored, and none of them can reach the target node, so this node is also unreachable.
- *isDone* = true and *isReachable* = true: This state means that all the neighbors of the node have been explored, and at least one of them can reach the target node, so this node is also reachable.

At the beginning of the parallel algorithm, we initialize some variables for the target node in lines 1–2. We set both *isDone* and *isReachable* flags of the target node to true, because it is the only reachable node at the start of the algorithm, and its reachability will be inherited by other nodes during backtracking. Moreover, when a thread reaches the target node, it does not need to explore further, so we use the done flag to terminate the exploration at this point. After pushing the start node into the stack, each thread starts a parallel depth-first search. In lines 7–8, at each iteration, we check whether the current node is done or not. If it is true, we pop it out of the stack promptly to avoid unnecessary traversal. In lines 10–14, we determine whether a node is done and reachable or not. A node becomes done when all its neighbours have been visited. Once a node is done, we can decide its reachability status. If any neighbour of a node is reachable, then all nodes in its SCC should be reachable as well. In lines 16–17, if a node still has unvisited neighbours, we

---

**Algorithm 3** Finding all PathNodes between *s* and *t* in a graph (parallel).

**Input:** The Graph stored in a Directed acyclic graph $G'$, The starting node *s*, The target node *t*

**Output:** Graph $G'$ with reachable markers

```
1   V(G').isDone ← false;
2   V(G').isReachable ← false;
3   t.reachable ← true;
4   t.isdone ← true;
5   stack ← ∅;
6   foreach thread do
7     stack.push(s);
8     while stack is not empty do
9         v ← stack.top;
10        if v is done then
11            stack.pop;
12        end
13        else
14            if N'(v) is all been traversed then
15                v.isdone ← true;
16                if N'(v) contain a reachable node then
17                    S(v).isreachable ← true;
18                end
19                stack.pop;
20            end
21            else
22                u ← N_random(v);
23                stack.push(u);
24            end
25        end
26    end
27  end
```

---

randomly select one representative neighbour to explore and complete the depth-first search.

In Algorithm 1, a traditional DFS is used to find all nodes between two specified nodes, including those within the cycles to which they belong. Assuming the graph comprises V nodes and E edges, where N represents the number of nodes and E represents the number of edges, in the worst-case scenario, the algorithm requires visiting every node and edge, resulting in a time complexity of O(N + E). Algorithm 2 adopts an approach that dynamically identifies and constructs SCCs during processing, without the need to

**Table 2** Statistic of the datasets.

| Dataset | Name | V | E |
|---|---|---:|---:|
| Iceland | IL | 75 | 114 |
| Political-Books | PB | 105 | 441 |
| Jazz-Musicians | JM | 198 | 2,742 |
| Wiki-Vote | WV | 7,115 | 103,609 |
| Soc-Epinions | SE | 75,879 | 508,837 |
| Email-EuAll | EE | 265,214 | 420,045 |
| Soc-Pokec-Relationship | SPR | 1,632,083 | 30,622,564 |

explicitly create the transitive closure of the entire graph. Generally, this algorithm exhibits linear time complexity. Algorithm 3 fundamentally employs a parallel DFS approach, with its complexity also being O(N + E), but it becomes more efficient than O(N + E) when executed in a parallel computing environment. Algorithm 2's transformation of the graph into a Directed Acyclic Graph, followed by its handover to Algorithm 3's parallel DFS for path detection, significantly enhances the performance of the algorithm.

## EVALUATION

In this section, we evaluate the efficiency of the proposed algorithms. We conducted our experiments on a computer with the following configuration: a 3.2 GHz Intel Core i9-12900K processor with a FLOPS performance of 1,180 GFLOPS, 16 GB of RAM, a 256 GB SSD. The operating system was Windows 10 Pro (64-bit). We used C and C++ for implementing our algorithm.

In our experimental evaluation, we employed a range of parameters to ensure the robustness and reliability of our results. We assessed the performance of our algorithms on seven real-world benchmark graph datasets, as detailed in Table 2. To analyze the impact of parallelism, we conducted experiments using varying numbers of threads, ranging from 1 to 6 threads. This allowed us to evaluate the efficiency of our algorithm under different multi-threaded configurations. To ensure result reliability, each experiment was run at least 10 times, and the averages were computed. This approach effectively minimizes the influence of random variations and provides a more accurate representation of our algorithm's performance. Concurrently, we recorded the time consumption and memory usage for each experiment, employing appropriate metrics to quantify the algorithm's efficiency and resource utilization.

In our implementation, we utilized standard computing libraries and tools commonly used in the fields of computer science and data analysis. We leveraged the thread library from the C++ Standard Library to facilitate multi-threading and parallel processing. Additionally, we utilized the mutex library to provide synchronization primitives for managing shared resource access in multi-threaded programs. For handling graph data structure operations and analysis, we relied on widely used C++ containers such as *unordered_map* and *unordered_set*.

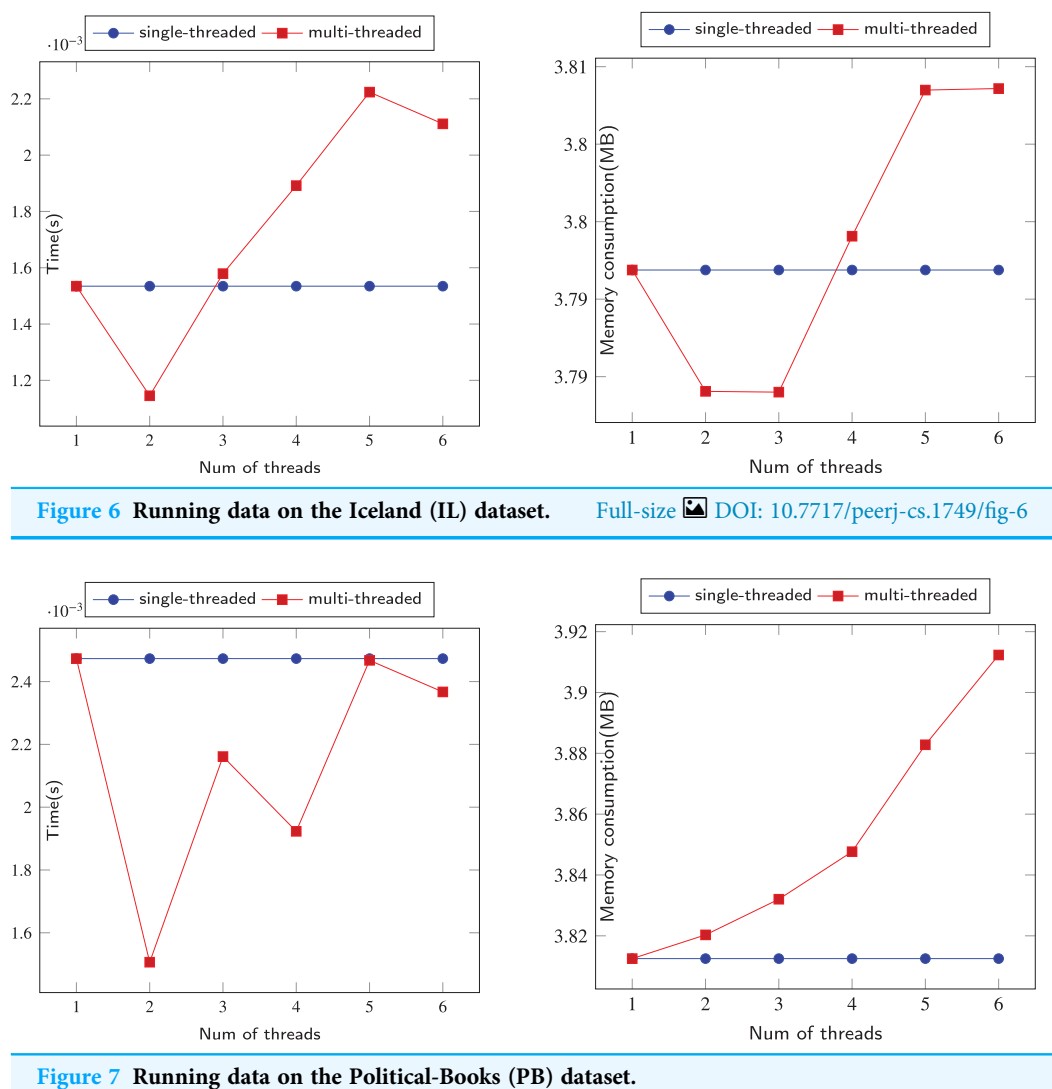

**Figure 6  Running data on the Iceland (IL) dataset.**

**Figure 7  Running data on the Political-Books (PB) dataset.**

The primary focus of our algorithm lies in the efficient identification of potential fraudulent bank accounts through the utilization of parallelized DFS on a meticulously optimized data structure. Our algorithm enhances parallelism by guiding each thread towards distinct portions of the graph through the random selection of neighboring nodes, while simultaneously achieving load balancing through collaborative updates of node states within a shared data structure. By introducing randomness and enabling communication of node states among threads, we have successfully reduced the execution time of our algorithm through parallelization.

## Performance

The comparison of sequential and parallel DFS is shown on Figs. 6–12. We tested our parallel depth-first search algorithm on seven datasets. These datasets spanned from tens to millions of vertices and from thousands to millions of edges. We measured the performance of our algorithm under different numbers of threads, running each

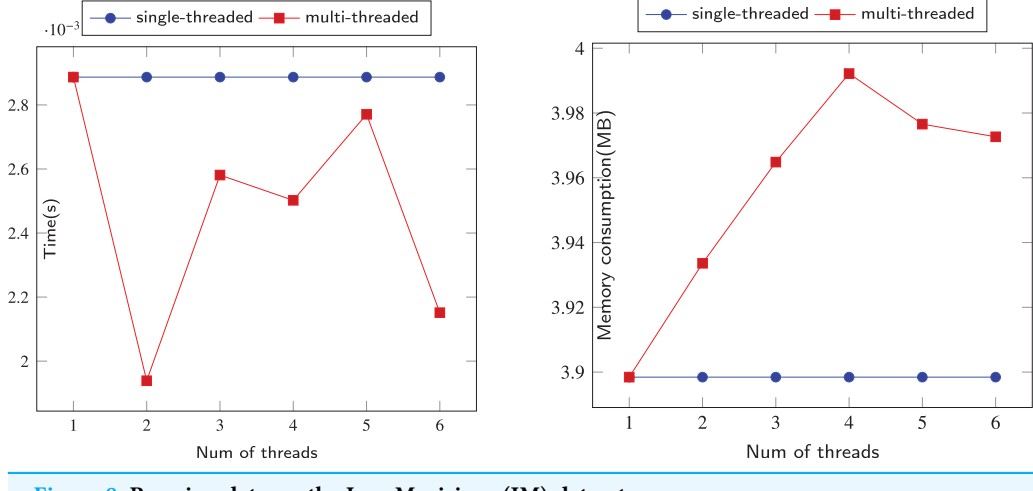

**Figure 8  Running data on the Jazz-Musicians (JM) dataset.**

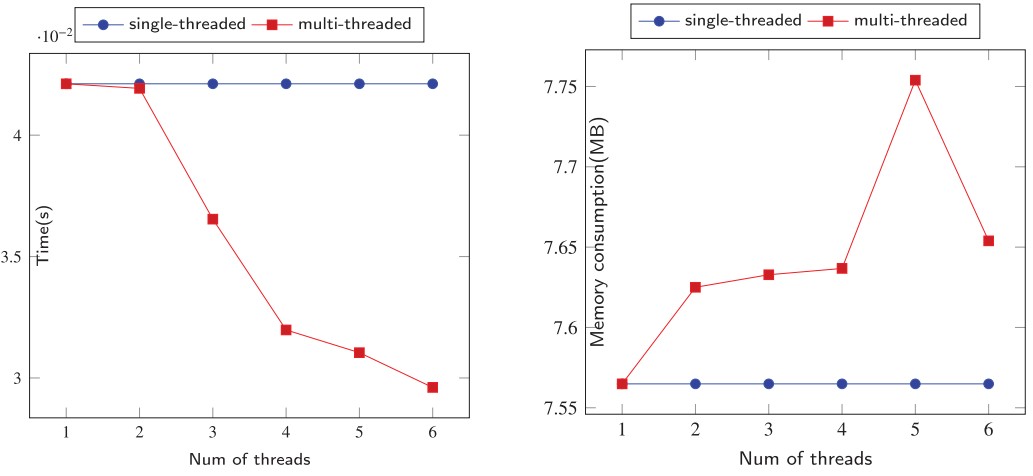

**Figure 9  Running data on the Wiki-Vote (WV) dataset.**

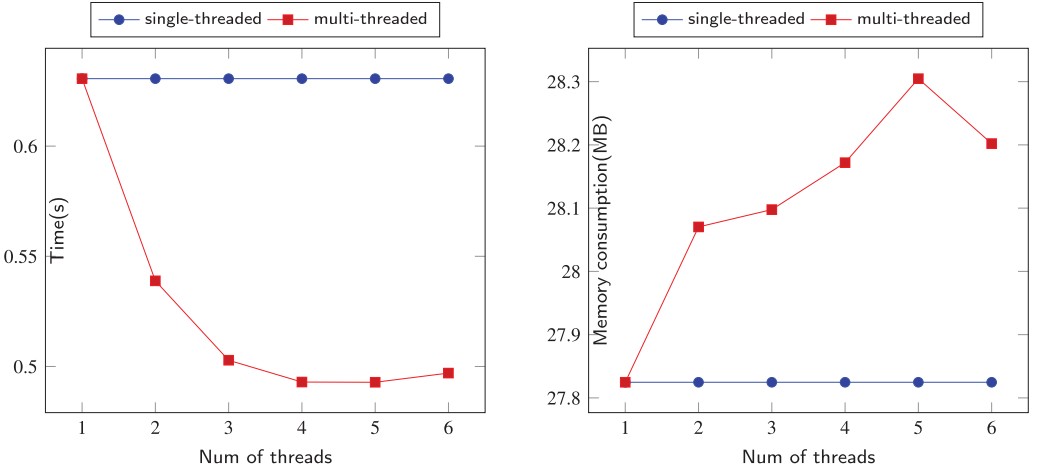

**Figure 10  Running data on Soc-Epinions (SE) dataset.**

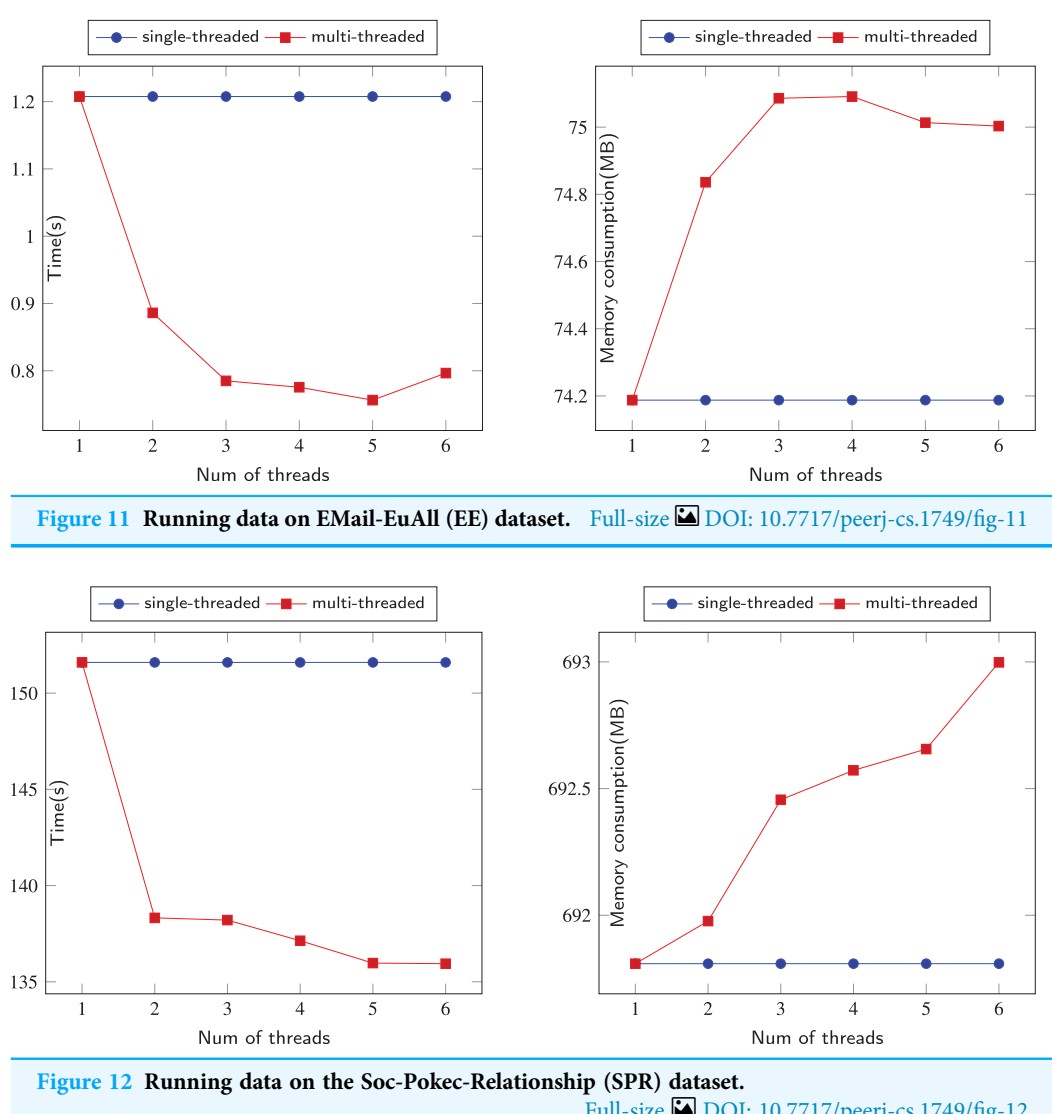

**Figure 11  Running data on EMail-EuAll (EE) dataset.**

**Figure 12  Running data on the Soc-Pokec-Relationship (SPR) dataset.**

experiment at least 10 times and using the average values to compute all results. We compared the time and memory consumption across different threads and discussed the trade-offs and challenges involved.

When we use dataset IL to test our algorithm, as shown in Fig. 6, when the number of threads is 2, the time consumed by the algorithm is better than that of a single thread. However, when the number of threads is greater than 2, the time consumed by the parallel algorithm exceeds that of a single thread, and there is a significant performance decline. This is largely because IL is a small-scale dataset with only 75 vertices and 114 edges, which means that each thread needs to build a stack to process this simple graph. This makes the resources consumed by parallelism greater than the performance advantages brought by parallelism. In terms of memory consumption, when the number of threads is small, threads can still share the pressure of other threads, but when the number of threads grows

to more than 3 for this small-scale dataset, multiple threads will only increase redundant calculations.

When the dataset is slightly larger, the advantages of our algorithm are reflected. Our algorithm's performance on the PB and JM datasets is shown in the Figs. 7 and 8. Due to the randomness of each thread selecting a neighboring node, the time consumption of our algorithm did not show a linear relationship with the number of threads, but fluctuated slightly. However, overall, our parallel depth-first search algorithm was more time-efficient than the serial algorithm, and achieved this with an acceptable memory consumption.

We ran our algorithm on the WV dataset, which is a relatively small dataset with about 7,000 nodes and more than 100,000 edges. As shown in Fig. 9, When the number of threads is 2, the time consumption of multi-threading is only slightly better than that of single-threading, but as the number of threads increases, the time consumption of the algorithm is much lower than that of single-threading, which indicates that multi-threading can better share the pressure of graph processing between each thread. In terms of memory consumption, multi-threading occupied more memory than single-threading, and also showed a substantial increase when the number of threads reached 5. This is because multiple threads may push the same nodes onto the stack, which adds extra computations and increases the memory burden of the program.

Our algorithm also performed well on the larger datasets SE and EE. As shown in Figs. 10 and 11, the time performance of our algorithm improved gradually with the increase of the number of threads on slightly larger datasets. The memory consumption also slowed down with more threads. With the help of multithreading, we achieved better results than the serial program.

We now turn to the performance of our algorithm on larger datasets SPR. The SPR dataset has more than 1.6 million vertices and more than 30.6 million edges, making it a challenging test case for our algorithm. as shown in the (Fig. 12), Our algorithm has also demonstrated the feasibility of scaling up to large datasets by utilizing multi-threading, while incurring reasonable additional memory overhead. This verifies that our algorithm can achieve better time performance than a single-threaded approach.

In Figs. 6–12, we observe that the memory consumption for the multi-threaded approach is indeed higher than that of the single-threaded approach. When detecting financial flows within bank accounts, the speed of detection is crucial for effectively identifying and preventing fraudulent activities. Therefore, we are willing to trade memory consumption for runtime efficiency. Our goal is to strike a balance that ensures a significant improvement in algorithm execution time while keeping memory consumption within manageable limits. Even with the largest dataset, SPR, the increase in memory usage due to the addition of threads remains within acceptable bounds, not exceeding 5 MB in total. This approach allows us to achieve the expected efficiency and effectiveness in identifying and addressing fraudulent activities.

In Table 3, we compared the time efficiency between DFS and our algorithm. DFS (*Grossi, Marino & Versari, 2018*; *Peng et al., 2019*; *Rizzi, Sacomoto & Sagot, 2014*) is a brute-force search algorithm that finds all nodes that may be passed from the start node to the target node by depth-first search. There are two approaches to solve the DFS problem:

**Table 3 Comparison of time consumption of different algorithms on different datasets.**

| Dataset (start → target ) | DFS | PathNodes | Our algorithm in serial | Our algorithm in parallel | | PathNodes |
|---|---|---|---|---|---|---|
| | | | | Minimum | Maximum | |
| IL(1→45) | 0.0012454 | 26 | 0.0015345 | 0.0011453 | 0.0022238 | 26 |
| PB(1→35) | 4.92856 | 33 | 0.0024732 | 0.0015063 | 0.0024676 | 33 |
| JM(112→137) | 48.89013 | 19 | 0.0028869 | 0.0019388 | 0.0025811 | 19 |
| WV(5→ 61) | INF | – | 0.0421159 | 0.296087 | 0.419268 | 1,302 |
| SE(0→4) | INF | – | 0.6306173 | 0.496998 | 0.538866 | 32,233 |
| EE(0→1) | INF | – | 1.2075687 | 0.756337 | 0.886058 | 34,203 |
| SPR(1→13) | INF | – | 151.59621 | 135.941559 | 138.321742 | 1,304,537 |

recursive and non-recursive. However, in the context of this experiment, the graph size is relatively large, which introduces the risk of stack overflow due to excessively deep recursive function calls when using recursive DFS. Therefore, we opted to compare the runtime of a non-recursive DFS approach that utilizes an explicit stack with parallel algorithms.

We have summarized the performance of DFS and our algorithm in Table 3. In this table, the *maximum* and *minimum* values for our algorithm in parallel represent the slowest and fastest execution times of our algorithm with different threads. In some datasets, the time consumption of the DFS algorithm is marked as *INF*, indicating that it cannot find suspicious nodes within the acceptable time range for the banking institution. Analyzing the table, it is evident that for the IL dataset, our algorithm only marginally outperforms DFS in the best-case scenario of parallel execution. However, in most cases, the time consumption of our algorithm is higher than that of DFS. This is attributed to the small size of the IL dataset, which consists of only 75 nodes and 114 edges. As a result, there is a higher probability of multiple threads randomly selecting the same nodes, leading to increased resource overhead for updating node states in parallel compared to the resources required for single-threaded execution. On the other hand, for moderate-sized datasets like PB and JM, the advantages of our algorithm gradually become apparent. Table 3 demonstrates that our algorithm consistently outperforms DFS in terms of runtime. This improvement can be attributed to the inherent randomness of our algorithm, which guides threads to different parts of the graph, effectively distributing the graph processing workload and significantly enhancing the algorithm's efficiency. For larger datasets such as WV, SE, EE, and SPR, our algorithm surpasses the traditional DFS algorithm by a significant margin in terms of runtime. The parallelization strategy employed in our algorithm proves highly effective when handling large graphs. The utilization of random selection and coordinated updates allows for optimal workload distribution among threads, resulting in enhanced efficiency.

## Algorithm validity

As our parallel DFS algorithm selects neighboring nodes in a random manner during the graph exploration process, each thread may explore different parts of the graph in each iteration. To ensure that there are no false positives or false negatives, we have established strict definitions for node states. To validate the effectiveness of our algorithm, we use the suspicious nodes detected by the traditional DFS algorithm as a benchmark. In Table 3, the third and seventh columns represent the number of suspicious accounts detected by the two algorithms, respectively. It can be observed that in the IL, PB, and JM datasets, we obtained a same number of suspicious nodes as traditional DFS. This indicates that our parallel algorithm can correctly detect fraudulent nodes while ignoring innocent ones.

## CONCLUSION

In this article, we proposed a novel parallel path detection algorithm for identifying suspicious fraudulent accounts in large-scale banking transaction graphs. Our algorithm is based on a three-step approach that involves constructing a directed graph, shrinking strongly connected components, and using a parallel depth-first search algorithm to mark potentially fraudulent accounts. We used Vincent Bloemen's multi-core on-the-fly SCC algorithm for finding all strongly connected components in linear time, which enabled us to handle large-scale graphs with exponential growth.

Our algorithm achieved high performance and scalability on multi-core processors, making it a promising solution for detecting suspicious accounts and preventing money laundering schemes in the banking industry. The evaluation results demonstrate that our algorithm outperforms serial time baselines in terms of both runtime and scalability. Furthermore, our approach can significantly improve the accuracy of financial regulation on bank accounts, enabling risk-labeling of anomalous accounts in transaction paths.

Overall, our work contributes to the ongoing efforts to combat financial fraud and promote financial stability in the banking sector. In the future, we plan to investigate the effectiveness of our algorithm in detecting other types of financial fraud, such as insider trading and market manipulation, to further improve the accuracy of fraudulent account detection.

### Funding

This work was funded by the Natural Science Foundation of Fujian Province (Grant No. 2021J01320), and the National Key Technology Research and Development Program of the Ministry of Science and Technology of China (Grant No. 2022YFB4300504). The funders had no role in study design, data collection and analysis, decision to publish, or preparation of the manuscript.

### Grant Disclosures

The following grant information was disclosed by the authors:
Natural Science Foundation of Fujian Province: 2021J01320.

National Key Technology Research and Development Program of the Ministry of Science and Technology of China: 2022YFB4300504.

## Competing Interests

The authors declare that they have no competing interests.

## Author Contributions

- Zuxi Chen conceived and designed the experiments, prepared figures and/or tables, authored or reviewed drafts of the article, and approved the final draft.
- ShiFan Zhang performed the computation work, prepared figures and/or tables, authored or reviewed drafts of the article, and approved the final draft.
- XianLi Zeng conceived and designed the experiments, prepared figures and/or tables, authored or reviewed drafts of the article, and approved the final draft.
- Meng Mei analyzed the data, prepared figures and/or tables, authored or reviewed drafts of the article, and approved the final draft.
- Xiangyu Luo performed the computation work, prepared figures and/or tables, authored or reviewed drafts of the article, and approved the final draft.
- Lixiao Zheng performed the experiments, prepared figures and/or tables, authored or reviewed drafts of the article, and approved the final draft.

## Data Availability

The data is available at Zenodo: Z, S. (2023). Parallel Path Detection Test Dataset [Data set]. Zenodo. https://doi.org/10.5281/zenodo.10047614.

The code is available at Zenodo: sofacoder35. (2023). sofacoder35/path_detection: Parallel Path Detection for Fraudulent Accounts in Banks (v1.0). Zenodo. https://doi.org/10.5281/zenodo.10047528.

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
