# Peer review of "Parallel path detection for fraudulent accounts in banks based on graph analysis"

_PeerJ Computer Science, doi:10.7717/peerj-cs.1749_

## Round 0.1 · original submission · Major Revisions

We regret to inform you that your manuscript has not been accepted for publication with its current form since the methods are not clear. We kindly ask you to revise your manuscript by considering the reviewers' comments. More details are needed.

Reviewer 1 ·

Basic reporting

This paper presents a novel parallel path detection algorithm for identifying suspicious fraudulent accounts in large-scale banking transaction graphs. Authors aim to detect suspicious accounts and preventing money laundering schemes in the banking industry.

There are several writing issues on the paper based on the reference format. It should be the same format. In addition, authors considers the spaces between references and text, for example in Line 35, however in Lines 26-27 spaces were not considered. Therefore, all the issues should be checked.

There are a lot of typos in the paper. Authors should review the paper carefully.

Experimental design

Adjacency list as the data structure for the directed graph was used to store nodes in the proposed method. It may be accepted for the seperated graph and given experiment; but if a graph will be dense, it may cause inefficent memory usage even if pointers were used. Therefore, authors should be given more explain this point.

Authors should give more details for the parallelism of the approach. Which is/are problem(s) to be solved by this approach?

The recursive DFS algorithm is already considers cycles. What is the novelty on this point? Authors mentions about it as a different way but it is already existing appraoch for the DFS.

Validity of the findings

What is the limitations for number of threat?

In the Fig.12, memory consumption for the multi-threaded approach increases depending on increasing of the number-of-thread. Furthermore, single-threaded approach provides fixed memory consumption. Authors declerated that it is a verification of running time performances of the proposed method. What is the success metric for this experiment? Can we say that the proposed approach provides efficiency only considering running time. Isn't the memory consumption a success metric?

Additional comments

Authors should provide more clarity for the given comments and the proposed approach should be explained more considering mentioned points.

Cite this review as

·

Basic reporting

The basic inspection of this article is good. It have been edited using clear, unambiguous professional English throughout. The introduction and literature review sections provide sufficient background of this manuscript. All references are relevant. The structure of this manuscript follows the PeerJ standard and discipline norm. All figures are relevant, high quality, well labeled and described.

Experimental design

The original primary research is within the scope of the journal. The research question is well defined by authors. This research question is relevant and meaningful in money laundering problems. Nevertheless, I suggest the authors state their proposed method may be employed to detect which money laundering typology. This manuscript proposes that this article was devoted to money laundering detection. A webpage lists some examples. Besides, although the authors have performed a rigorous investigation in comparing the execution time of their algorithms, I suggest the authors further define their accuracy standard. Is it possible that the author’s’ algorithms filter wrong vertices? The authors didn’t provide similar investigations.

Validity of the findings

The authors have not addressed the impacts and novelty of their article. I suggest the authors may state that their algorithms can replace some of auditors’ works in searching fraudulent financial accounts. Besides, the authors have described statistically data employed to implement their study. Unfortunately, these data may not be related to the main interest (i.e. money laundering) of this article. But, open data satisfying this main interest may be rare. Therefore, the data employed to edit this article are acceptable. The conclusion of this article are well stated, linked to original research questions and limited to supporting results.

Additional comments

Overall, I suggest accepting this article after a minor revision. The authors need to clarify something for accepting this article.

Reviewer 3 ·

Basic reporting

In the "Related Work" section, it is recommended that authors incorporate a table detailing the computational limitations of key state-of-the-art methodologies.

Experimental design

Authors are required to include a concise discussion concerning the simulation parameters and computational libraries employed, specifically within the Experimental Evaluation section.

Validity of the findings

Authors are advised to dedicate a paragraph to the discussion of the time complexity associated with Algorithm 1.

Additional comments

None.

Cite this review as

·

Basic reporting

All the references are relevant but do not cover the appropriate key research. The references are old. The authors must include more recent references to cover appropriate key research.

Experimental design

no comment

Validity of the findings

no comment

Additional comments

no comment

---

## Round 0.2 · Minor Revisions

The paper needs a minor revision. Please check the reviewers' comments and revise the paper accordingly.

Reviewer 1 ·

Basic reporting

This paper presents a novel parallel path detection algorithm for identifying suspicious fraudulent accounts in large-scale banking transaction graphs.

The paper has been improved according to the reviews received in the previous rounds and is now fluent and interesting.

There are still some typos in the paper so, authors should review the paper in terms of writing.

Experimental design

Authors use the DFS algorithm and evaluate the results in terms of running time. DFS algorithm with recursion, it may be run at more less time. Did the authors simulate it? If not, it must be a reason behind it.

Validity of the findings

No comments.

Cite this review as

·

Basic reporting

This article presented sufficient field background. The authors used a professional article structure, figures, and tables and shared raw data.

Experimental design

The research question in this manuscript was well-defined, relevant, and meaningful. It presented a faster algorithm for visiting vertices in a graph structure. This algorithm is helpful for detecting nodes simulating fraudulent financial accounts. Nevertheless, readers of this article may desire to know whether the proposed algorithm can correctly detect fraudulent nodes but passe innocent ones. Such a desire was not mentioned in this article. The authors may briefly state whether their works can complete the desire.

Validity of the findings

All underlying data have been provided; they are robust, statistically sound, & controlled. The data on which the conclusions are based are available in an acceptable discipline-specific repository. These data are robust, statistically sound, and controlled. Conclusions are well stated, linked to the original research question & limited to supporting results. Those conclusions were appropriately stated and connected to the original question in this article.

Additional comments

The only question of this article is the experimental design. In addition to the fast visiting of all nodes in a graph, we want to filter out susceptible nodes.

Reviewer 3 ·

Basic reporting

Authors have implemented the suggestions in the revised version of the manuscript.

Experimental design

Authors must provide the following information in the experimental evaluation segment of the manuscript.

- The load balancing approach and work distribution strategy that were adapted for the experiment.

- Detailed FLOPS (Floating Point Operations Per Second) information related to the machine where the experiments were conducted, to give a clear perspective on computational capabilities.

- A comprehensive analysis of the speed-up achieved, indicating the efficiency and effectiveness of the implemented parallel solution.

Validity of the findings

Authors have incorporated the suggested revisions.

Additional comments

None.

Cite this review as

---

## Round 0.3 · accepted · Accept

The reviewers' comments have been addressed. We happy to inform you that your manuscript has been accepted for publication.

Reviewer 1 ·

Basic reporting

The authors have incorporated all of my suggestions, and the manuscript can be accepted for publication.

Experimental design

The authors have incorporated all of my suggestions, and the manuscript can be accepted for publication.

Validity of the findings

The authors have incorporated all of my suggestions, and the manuscript can be accepted for publication.

Additional comments

None

Cite this review as

·

Basic reporting

Clear and unambiguous, professional English used throughout.

Experimental design

Original primary research within the Aims and Scope of the journal.

Validity of the findings

Conclusions are well stated, linked to the original research question & limited to supporting results.

Additional comments

The review of this article can end.